# Exploring the Impact of COVID-19 on Social Life by Deep Learning

**Jose Antonio Jijon-Vorbeck and Isabel Segura-Bedmar \***

Computer Science Department, Universidad Carlos III de Madrid, Avenida Universidad, 28911 Madrid, Spain;
ja.jijon.vorbeck@gmail.com
**\*** Correspondence: isegura@inf.uc3m.es

**Abstract:** Due to the globalisation of the COVID-19 pandemic, and the expansion of social media as the main source of information for many people, there have been a great variety of different reactions surrounding the topic. The World Health Organization (WHO) announced in December 2020 that they were currently fighting an "infodemic" in the same way as they were fighting the pandemic. An "infodemic" relates to the spread of information that is not controlled or filtered, and can have a negative impact on society. If not managed properly, an aggressive or negative tweet can be very harmful and misleading among its recipients. Therefore, authorities at WHO have called for action and asked the academic and scientific community to develop tools for managing the infodemic by the use of digital technologies and data science. The goal of this study is to develop and apply natural language processing models using deep learning to classify a collection of tweets that refer to the COVID-19 pandemic. Several simpler and widely used models are applied first and serve as a benchmark for deep learning methods, such as Long Short-Term Memory (LSTM) and Bidirectional Encoder Representations from Transformers (BERT). The results of the experiments show that the deep learning models outperform the traditional machine learning algorithms. The best approach is the BERT-based model.

**Keywords:** natural language processing; sentiment analysis; multi-classification; machine learning; deep learning; COVID-19; Long Short-Term Memory; LSTM; v; BERT

## 1. Introduction

The massive spread of the COVID-19 virus throughout the world in a very fast and uncontrolled manner has resulted in one of the most difficult crises that our society has faced in the last several decades. In July 2021, the total number of deaths related to COVID-19 worldwide surpassed 4 million. Moreover, COVID-19 has led to the most serious economic crisis in a century.

However, the impact of the COVID-19 pandemic has not only affected our health and economy, but it has also significantly affected other aspects of human life. Never before in the history of humankind has information been transferred so rapidly and reached so many people during a pandemic or any other kind of event [1]. The internet has made it possible to reach any country in the world in a matter of seconds, and social media has connected millions of people, who can express their feelings, concerns and thoughts with just a few clicks. Social media platforms such as Facebook and Twitter can be useful tools to communicate between family, friends and other groups of people, but they can also be used as a tool to spread misinformation and hate. As González-Padilla and Tortolero-Blanco mention, the worst aspect of social media is the potential to disseminate erroneous, alarmist and exaggerated information that can cause fear, stress, depression and anxiety in people with or without underlying psychiatric illnesses.

This is where natural language processing (NLP) comes into play. Several NLP applications can help to detect erroneous, alarmist and hateful messages. Besides this, NLP

can also identify users' emotions regarding a given topic—for example, the pandemic. These NLP techniques can analyse millions of tweets within a very short time, preventing the spread of this kind of information and finding solutions to deal with the infodemic. Moreover, both governments and public health organisations are some of the most benefiting actors, since, with the appropriate application of NLP, they will be able to better decide where, how and when to tackle new social problems before they arise and become more difficult to manage. For example, depression or anxiety problems among youth populations might be detected early on, and some strategies can be applied to balance and come to solutions early on in the process. Thus, NLP technology can be a new strategy to detect and prevent mental health problems among the population [2].

The main contribution of this study is to make a comparison between different machine learning and deep learning models to perform sentiment analysis on a collection of tweets about the COVID-19 pandemic. The tweets were gathered for a data science competition in Kaggle, Coronavirus tweets NLP —Text Classification [3] . The tweets were classified into five different sentiment categories (from extremely negative to extremely positive), and the whole collection was divided into training and test datasets. Thus, we performed a finer-grained sentiment analysis of tweets about the COVID-19 pandemic than previous works. We explored several traditional machine learning algorithms and some more advanced deep learning techniques, such as Long Short-Term Memory (LSTM) [4] and Bidirectional Encoder Representations from Transformers (BERT) [5]. These deep learning models have been proven to outperform the classical machine learning approaches for many NLP tasks [5].

## 2. Materials and Methods

### 2.1. Dataset

The dataset consisted of a collection of tweets gathered for the purpose of a competition in Kaggle, Coronavirus tweets NLP-Text Classification, https://www.kaggle.com/datatattle/covid-19-nlp-text-classification/activity (accessed on 3 November 2021). It was manually tagged by a team of linguistics experts. Moreover, all private names and surnames were removed in order to maintain data privacy.

The tweets were manually annotated into five different classes: extremely positive, positive, neutral, negative and extremely negative. The task of annotating the tweets with a sentiment was always a subjective decision. In particular, distinguishing between negative and extremely negative (or positive and extremely positive) could be very difficult. For example, the tweet " *Yesterday no chicken breast on the truck today no redmeat. #HoardersGonnaHoard ??*" was annotated as negative, but it could also have been annotated as extremely negative.

The dataset was then split in two different sets: the training dataset, with 41,156 tweets, and the test datasets, with 3798 tweets.

Figures 1 and 2 show the class distribution of the training and test datasets, respectively. All the classes were balanced in both datasets.

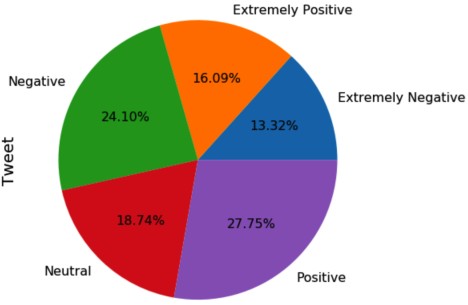

**Figure 1.** Class distribution of the training dataset.

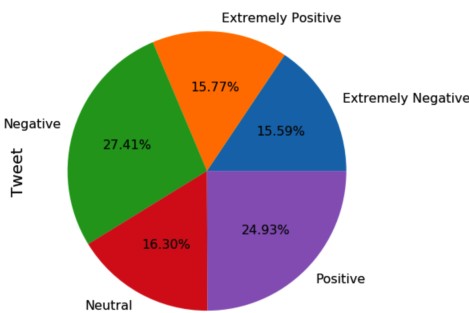

**Figure 2.** Class distribution of the test dataset.

Some prepossessing tasks were applied to the input tweets in order to allow them to be processed more practically by the algorithms. Firstly, cleaning of the tweets using regular expressions to remove hashtags, mentions and web page links was performed. These items were removed because they did not provide a proper semantic relation with the rest of the words in the text.

*2.2. Traditional Machine Learning Algorithms*

First, we proposed several traditional machine learning algorithms, such as Support Vector Machine (SVM), Logistic Regression, Gaussian and Multinomial Naive Bayes and Random Forest, which have been successfully used for the task of text classification.

All the classical machine learning models were developed using the Python Sklearn library [6], which allows the performance of Hyper Parameter Optimisation (HPO).

Optimising the hyperparameters is essential to obtain good performance. The different options of hyperparameters could have a great impact in the evaluation of the model [7]. Therefore, techniques such as grid search or randomised search for hyperparameter tuning are very common and have been used by many researchers [8]. During this experiment, both approaches were used to find the best hyperparameters for the classical machine learning algorithms.

All the hyperparameters for the models are presented in Table 1. For this purpose, both the randomised and grid search were used. In the case of the Logistic Regression model, the randomised search took more than 12 h to run and did not converge to a solution. Therefore, we only used grid search for this classifier.

**Table 1.** Hyperparameters selected per model.

| Model | HyperParameters | HPO Method |
|-------|-----------------|------------|
| SVN | {C = 1, kernel = 'rbf', tol = $1 \times 10^{-3}$ and gamma = 'scale'} | Randomised Search |
| Logistic Regression | { C = 25, penalty = 'l2', tol = $1 \times 10^{-4}$, solver = 'lbfgs} | Grid Search |
| Gaussian Naive Bayes | {proirs = 5, var_smoothing = $1 \times 10^{-9}$ } | Randomised Search |
| Multinomial Naive Bayes | {alpha = 1, fit_prior = True, fit_classes = 5} | Randomised Search |
| Random Forest | {min_samples_split: 100, min_samples_leaf: 2, max_depth: 256} | Randomised Search |

*2.3. Deep Learning Models*

We now present the three different deep learning approaches used in this work: MultiLayer Perceptron (MLP), LSTM and BERT.

An MLP is a class of feed-forward artificial neural network that consists of at least three layers of nodes: one input layer, one hidden layer and one output layer. It uses the back-propagation training algorithm [9]. This consists of running the gradients of the obtained forward bass backwards in the chain of the "neurons", thus obtaining the error terms and then tuning their weights to obtain a lower loss [10]. MLP can be used as a benchmark for deep learning models.

Figure 3 provides information about the MLP architecture used in this work. It can be seen that an embedding layer was used at the beginning, followed by an average pooling layer and, finally, two dense layers with eight and five shapes were used. This architecture was based on some previous research dealing with the optimal architectures of NN [11]. The output layer used a "softmax" activation function, providing a probability output for each of the output neurons, each representing a sentiment class. This layer needed to have the same number of neurons as the number of classes in the problem—that is, five neurons. A total of over one million parameters were trained in this model, since most of them were derived from the embedding layer, which was a complex layer in itself, and the rest came from the dense layers.

```
Model: "sequential"
_________________________________________________________________
Layer (type)                 Output Shape              Param #
=================================================================
embedding (Embedding)        (None, 35, 128)           1280000
_________________________________________________________________
global_average_pooling1d (Gl (None, 128)               0
_________________________________________________________________
dense (Dense)                (None, 8)                 1032
_________________________________________________________________
dense_1 (Dense)              (None, 5)                 45
=================================================================
Total params: 1,281,077
Trainable params: 1,281,077
Non-trainable params: 0
_________________________________________________________________
```

**Figure 3.** Hyperparameters of MLP .

Our second deep learning architecture was LSTM, which is part of the family of Recurrent Neural Networks (RNN) [12]. Classical RNN might be difficult to train using the back-propagation algorithm used in simpler networks, because of the vanishing gradient problem [13]. This problem consists of the back-propagation error growing or decaying exponentially, thus impeding the weights from updating in an optimal manner [13]. LSTM avoids this problem by using gates that regulate the flow of information in the network. In particular, each LSTM cell has three gates: an input gate to read the relevant information, a forget gate to remove the non-relevant information and an output gate that produces the information for the next LSTM cell. This gating mechanism also allows the capture of long-term dependencies.

Figure 4 shows the different layers used in our LSTM model. As in the previous model, the first layer was the embedding layer, where each token was represented by a word embedding. Then, we used a bidirectional layer of LSTM, which allowed us to read the input sequence from left to right and from right to left. Some extra dropout layers and dense layers were added to the model. A total of more than 1.5 million parameters were tuned in this model. It is worth mentioning that the tuning of the different layers and parameters is an important topic in itself, which is beyond the scope of this work. Our LSTM model was trained for 20 epochs.

```
Model: "sequential_3"

_________________________________________________________________
Layer (type)                 Output Shape              Param #
=================================================================
embedding_3 (Embedding)      (None, 35, 128)           1280000
_________________________________________________________________
bidirectional (Bidirectional (None, 256)               263168
_________________________________________________________________
dense_7 (Dense)              (None, 128)               32896
_________________________________________________________________
dropout_1 (Dropout)          (None, 128)               0
_________________________________________________________________
dense_8 (Dense)              (None, 64)                8256
_________________________________________________________________
dropout_2 (Dropout)          (None, 64)                0
_________________________________________________________________
dense_9 (Dense)              (None, 32)                2080
_________________________________________________________________
dense_10 (Dense)             (None, 5)                 165
=================================================================
Total params: 1,586,565
Trainable params: 1,586,565
Non-trainable params: 0
_________________________________________________________________
```

**Figure 4.** Architecture of our LSTM model.

Our third deep learning model was BERT, which was published by a group of engineers working at Google in 2018 [5]. BERT is designed to pre-train deep bidirectional representations from unlabelled text by jointly conditioning on both the left and right context in all layers. This model was pre-trained with all the content from BookCorpus and Wikipedia, which together reach more than 3200 million words. To learn these representations, approximately 15% of the words were masked, and the model had to infer them from their corresponding contexts. The input of the model needed to be formatted in a particular way, which consisted of a word embedding for each token, a position embedding representing its location in the sentence and the segment embedding, which identified the sentence of the word. In addition, a mask embedding was used to represent whether the token was masked or not.

Figure 5 shows the different layers used to build our BERT model. After the input layers, we had the BERT layer (Keras layer), which learned a representation for each input token. Finally, a dense layer with five neurons was used as the output layer. A total of 109 million parameters formed part of the model, and this is one hundred times more than before. Our BERT model was trained for 20 epochs.

```
Model: "model"

__________________________________________________________________________________________________
Layer (type)                 Output Shape         Param #     Connected to
==================================================================================================
input_word_ids (InputLayer)  [(None, 160)]        0
__________________________________________________________________________________________________
input_mask (InputLayer)      [(None, 160)]        0
__________________________________________________________________________________________________
segment_ids (InputLayer)     [(None, 160)]        0
__________________________________________________________________________________________________
keras_layer (KerasLayer)     [(None, 768), (None, 109482241   input_word_ids[0][0]
                                                              input_mask[0][0]
                                                              segment_ids[0][0]
__________________________________________________________________________________________________
tf.__operators__.getitem (Slici (None, 768)       0           keras_layer[0][1]
__________________________________________________________________________________________________
dense (Dense)                (None, 5)            3845        tf.__operators__.getitem[0][0]
==================================================================================================
Total params: 109,486,086
Trainable params: 109,486,085
Non-trainable params: 1
__________________________________________________________________________________________________
```

**Figure 5.** Architecture of our BERT model .

For training the deep learning models, we used the Google Colaboratory Environment (Google Colab) [14], since it provides a free virtual GPU machine. The models were implemented by using the library *Keras* [15]. This was chosen because of its simplicity of use, and because it runs on top of *TensorFlow*.

To make our experiments easily reproducible for the NLP community, we have created a public GitHub repository available at https://github.com/Joseantonio-96/Sentiment_ Analysis, accessed on 3 November 2021, with the code of all our models.

## 3. Evaluation

### 3.1. Metrics

First, it is necessary to describe the binary classification. In a binary classification problem, where there are only two cases, the confusion matrix resembles the example shown in Figure 6.

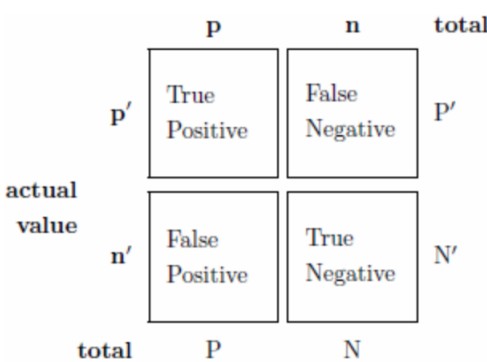

**Figure 6.** Binary classification confusion matrix.

Overall, we can only have four outcomes, which are briefly explained below.

- True Positive (TP): when the algorithm prediction matches the actual class of the instance, which is positive.
- True Negatives (TN): when the algorithm prediction matches the actual class of the instance, it being negative.
- False Positives (FP): when the prediction indicates that the instance is a positive when it truly is a negative.
- False Negatives (FN): when the prediction indicates that the instance is a negative when it truly is a positive.

The assessment of an algorithm in machine learning is usually done using the following metrics: precision, recall, F1-score and accuracy [16].

#### 3.1.1. Accuracy

The accuracy of a model measures the number of instances that were correctly classified, whether negative or positive cases. The equation below describes the accuracy in mathematical terms.

$$\text{Accuracy} = \frac{TP + TN}{TP + TN + FP + FN} \tag{1}$$

Accuracy is one of the metrics most used for classification tasks; it is not suitable for unbalanced datasets [17]. Achieving very high accuracy scores for a highly skewed dataset, when one of the classes might be less than one percent of the whole, is not meaningful at all. In our case, this was not a problem, since our dataset was balanced.

#### 3.1.2. Precision

Precision determines how many of the instances that are classified as positive are truly positive. This measure is given by Equation (2).

$$\text{Presicion} = \frac{TP}{TP + FP} \tag{2}$$

A model with a very high precision score would mean that the number of false positive cases is very low.

### 3.1.3. Recall

Recall, on the other side, determines the proportion of the instances classified as positive that are actually positive. The false negatives are taken into consideration in this metric.

$$\text{Recall} = \frac{TP}{TP + FN} \tag{3}$$

In this case, a model with a very high recall score would contain almost zero false negative values.

### 3.1.4. F1-Score

As a model that has a high precision score tends to have a low recall score, and vice-versa, the F1 is a combination of both metrics, used to obtain a unique metric through which to compare both aspects of the model. Mathematically, it is the harmonic mean between the recall and the precision. Its formula is given below:

$$\text{F1} = 2 * \frac{precision * recall}{precision + recall} \tag{4}$$

These metrics have been shown based on binary classification, but can be easily expanded to the multi-classification problem. Each of the classes will have its own precision, recall and F1.

Combining the different metrics for all the classes into a single number can be done using the macro-, micro- and the weighted averages. The macro-average is computed as the arithmetic mean of the individual scores of every class, while the weighted average takes into account the number of instances per class with respect to the total number of instances [18].

The micro-averaging metrics are used when each instance or prediction must be measured equally, while the macro-averaging metrics are used when the classes are to be treated equally, and they measure the performance of the classifier with regard to the most frequent class observed. Equations (5) and (6) show the mathematical expression of each of these averaging techniques, with the example of the precision score.

$$Precision_{Micro-average} = \frac{TP_1 + TP + 2 + \cdots + TP_n}{TP_1 + TP + 2 \ldots TP_n + FP_1 + FP_2 + \cdots + FP_n} \tag{5}$$

$$Precision_{Macro-average} = \frac{Precision_1 + Precision_2 + \cdots + Precision_n}{n} \tag{6}$$

where $n$ is the number of classes existing in the dataset.

The sklearn library [6] can be used to easily obtain all of these scores. A more detailed description of these metrics can be found in [16].

A suitable combination of these metrics must be used in order to select the best model. For sentiment analysis, the most common metrics are the accuracy and F1, with the latter being the most popular [19]. In our problem, we could consider accuracy as the reference score to compare the models because our dataset was balanced (see Figures 1 and 2).

### 3.2. Results

Table 2 shows a summary of the macro-average results and the accuracy of all the models. At first glance, it is easy to observe that the deep learning models all performed better than the classical machine learning ones. The BERT model obtained the highest scores, with an F1 of 71%, which is more than double that of the Gaussian Naive Bayes model.

**Table 2.** Macro-averaged results for each model. Best scores are in bold.

| Model | Precision | Recall | F1 | Accuracy |
|---|---|---|---|---|
| Gaussian Naive Bayes | 0.36 | 0.42 | 0.32 | 0.35 |
| Multinomial Naive Bayes | 0.52 | 0.42 | 0.43 | 0.45 |
| Random Forest | 0.49 | 0.47 | 0.47 | 0.47 |
| Support Vector Machine | 0.54 | 0.55 | 0.55 | 0.53 |
| Logistic Regression Model | 0.56 | 0.57 | 0.56 | 0.55 |
| Multilayer Perceptron | 0.66 | 0.66 | 0.66 | 0.65 |
| LSTM | 0.68 | 0.67 | 0.67 | 0.67 |
| BERT | **0.72** | **0.72** | **0.71** | **0.71** |

As there were five different classes in the study, a dummy classifier, which would assign all predictions to the greater class, would achieve a baseline score of 20% if the dataset was perfectly balanced. In our case, it would be closer to 25%. Thus, this must be taken into account as the lowest possible accuracy rate. A model with an accuracy value lower than this would perform very poorly. Therefore, it was necessary to use this as the lowest possible benchmark that we could achieve.

The model with the lowest accuracy was the Gaussian Naive Bayes. This model only reached an accuracy of 35%, with similar levels of precision and recall, but a very low F1-score of 35%. Then, we observed a significant jump in performance to reach the Multinomial Naive Bayes model, with an accuracy of 45% and similar values for the other metrics. It could be seen that the two probabilistic models were the lowest-performing models. This was because the size of the dataset, with the different classes and the non-normal distribution of the vectorized words, caused the probabilistic models to perform worse than the deterministic ones. The Random Forest model was next on the list, but not by a large margin, as its overall accuracy was only 47%.

SVM and Logistic Regression obtained accuracy values of 55% and 56%, respectively. Both classifiers provided the same F1-score of 55% over the test dataset. They were the best traditional machine learning classifiers for this task. In fact, previous works have proven that Logistic Regression usually achieves the best results in multiclass text classification tasks [20]. As could be seen, their performances were not great, and only 5 out of 10 tweets would have been correctly classified in their sentiment.

The top tier of models that were analysed in this study corresponded to the neural network models. Thus, MLP had an F1-score of 65%, 10 points greater than SVM and Logistic Regression. Its accuracy was also 10 points greater than Logistic Regression.

We now focus on the deep learning models. The LSTM model obtained an accuracy and an F1-score of 67%. In other words, LSTM provided slightly better results than MLP and significantly overcame the traditional machine learning classifiers. Finally, BERT achieved the best results, reaching an accuracy and an F1-score of 71%. BERT obtained improvements of 16% in accuracy and 15% in the F1-score over Logistic Regression, the best of the traditional machine learning models. BERT's scores were also significantly better than those obtained by the MLP, with a difference of 6 points above the accuracy of MLP (65%). When comparing the deep learning methods, it can be said that BERT showed an improvement of 4 points in terms of accuracy and F1-score.

Table 3 shows the classification scores of the logistic model divided by the sentiment classes. Both the negative and the positive class were the ones with the lowest F1-scores, with 45% and 53%, respectively. These instances might have been wrongly classified in the extremely negative or positive categories, since they could be difficult to distinguish.

**Table 3.** Results per class obtained by the logistic model.

| Class | Precision | Recall | F1-Score | Support |
|---|---|---|---|---|
| Extremely Negative | 0.56 | 0.56 | 0.56 | 592 |
| Negative | 0.50 | 0.41 | 0.45 | 1041 |
| Neutral | 0.59 | 0.69 | 0.64 | 616 |
| Positive | 0.50 | 0.56 | 0.53 | 947 |
| Extremely Positive | 0.66 | 0.63 | 0.64 | 599 |

LSTM has a similar behaviour (see Table 4), but the classes with the lowest performance were the negative and the extremely negative classes. A large increase in the neutral class was seen, and the positive class remained similar to the negative class in terms of F1-score.

**Table 4.** Results per class of the LSTM model.

| Class | Precision | Recall | F1-Score | Support |
|---|---|---|---|---|
| Extremely Negative | 0.55 | 0.72 | 0.62 | 592 |
| Negative | 0.66 | 0.60 | 0.63 | 1041 |
| Neutral | 0.80 | 0.72 | 0.76 | 616 |
| Positive | 0.62 | 0.68 | 0.65 | 947 |
| Extremely Positive | 0.79 | 0.63 | 0.70 | 599 |

Table 5 shows a more detailed evaluation for the BERT model, providing scores for each class. It can be seen that the precision scores for the negative and the extremely negative classes were much lower than those of the positive or the neutral ones. In terms of the F1-score, the worst-performing classes were the positive and the negative, with a value of 68% each. Compared to the neutral class, this was very poor, since the neutral class had an F1-score of 78%, ten points higher. Overall, the accuracy of the model was 71%, which, at first glance, does not appear particularly high, but dealing with multiple classes is not as simple as the binary classification problem, where the benchmark is at 50%. As mentioned before, in this study, having five categories, a dummy classifier would obtain a 20% accuracy score. Therefore, achieving more than 70% shows very good performance indeed.

Finally, we can conclude that BERT showed its clear superiority in this task and was clearly the most accurate model to use in this sentiment classification task.

**Table 5.** Evaluation scores per class for the BERT model.

| Class | Precision | Recall | F1-Score | Support |
|---|---|---|---|---|
| Extremely Negative | 0.66 | 0.76 | 0.71 | 592 |
| Negative | 0.67 | 0.68 | 0.68 | 1041 |
| Neutral | 0.81 | 0.76 | 0.78 | 616 |
| Positive | 0.69 | 0.67 | 0.68 | 947 |
| Extremely Positive | 0.79 | 0.73 | 0.76 | 599 |

All the previous systems [21–23] trained on the same Kaggle dataset [3] only addressed the task of classifying negative, neutral and positive messages. In other words, they merged the instances from the extremely negative and negative classes, and they did the same for the extremely positive and positive classes. Therefore, a direct comparison to these systems'

models cannot be done. Similarly, the study presented in [24] only detected negative, neutral and positive tweets, obtaining a very high F1-score (around 98%). However, its results cannot be directly compared to ours, because we performed a finer-grained sentiment analysis of tweets about the COVID-19 pandemic.

### 3.3. Confusion Matrices

Confusion matrices are an excellent visual tool that allow the observer to distinguish the classes in which a model performs better [25]. Figure 7 shows the confusion matrix created with the predictions provided by the Logistic Regression model. In this case, we can see that the model was not very accurate, especially on the negative side, as there was much more distortion in this part of the model. It could also be noted that the neutral class might have been the most complicated class to choose. The main diagonal around this class was more disperse, as values predicted as neutral might actually have been negative or positive, or vice versa. In fact, this is a very difficult problem, even for humans, since, sometimes, correctly labelling a tweet as neutral or not might be a very challenging task. It is also very subjective, and since the tweets were all annotated by different people, their different perceptions might have affected the results.

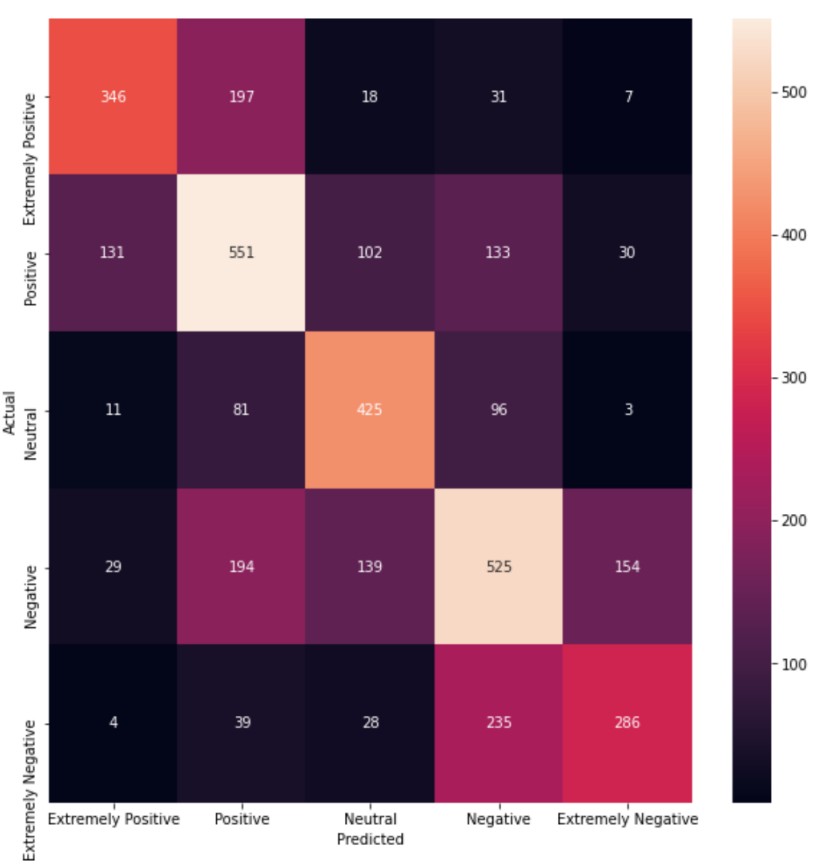

**Figure 7.** Confusion matrix for the logistic model predictions.

The confusion matrices for the rest of the classical machine learning algorithms can be easily obtained by using the notebooks https://github.com/Joseantonio-96/Sentiment_Analysis (accessed on 3 November 2021).

Continuing with the analysis of the results, the confusion matrices of the LSTM and BERT models are presented in Figures 8 and 9 , respectively. It is easy to see the differences in these matrices compared to the matrix in Figure 7. The main diagonals are much more highlighted than in the matrix of Logistic Regression, meaning that the predicted categories of the instances matched their actual values with greater frequency.

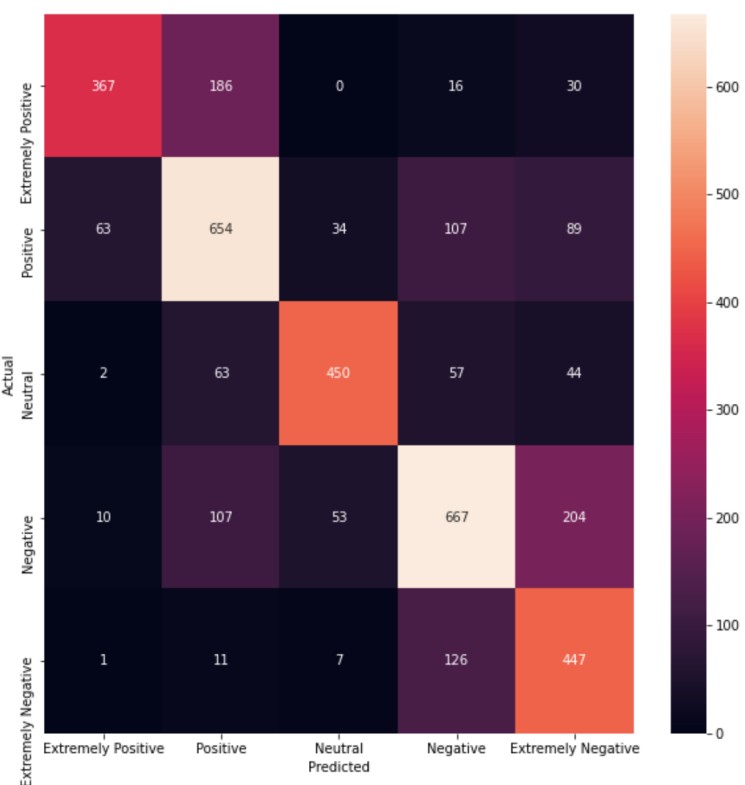

**Figure 8.** LSTM model.

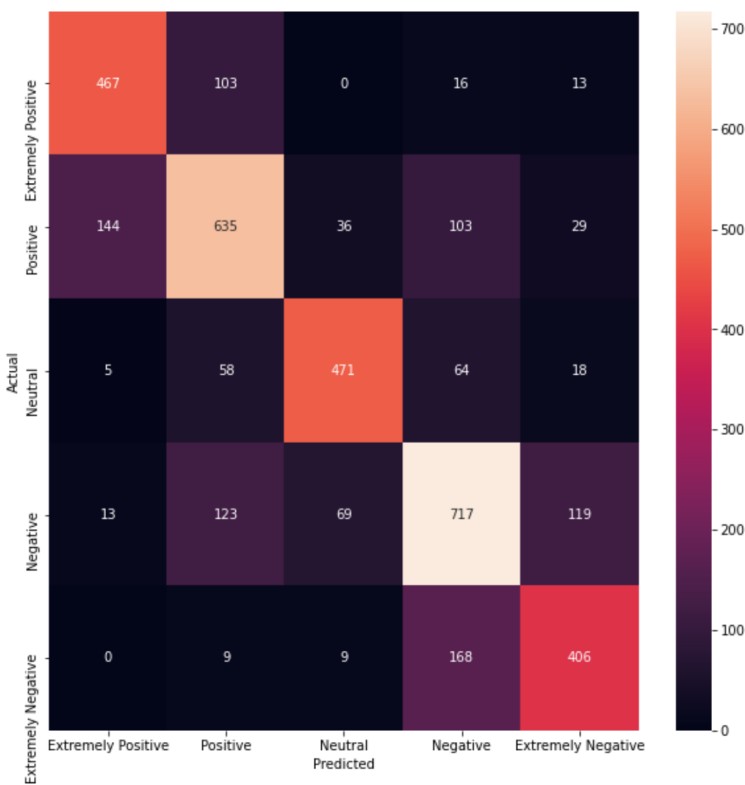

**Figure 9.** BERT model.

The main issue in the predictions of these two models was no longer the neutral class, but the difference between the extremely positive or extremely negative with the positive and the negative classes, respectively. The reader might note that there is a connection

between the misclassified observations in the four categories. Although the BERT model improved the precision of the positive polarities, the negative polarities still remained an issue. In fact, the difference between the extremely negative and negative class might be very different coming from two distinct persons, since it is a very subjective borderline.

### 3.4. Performance of the Models

The computational complexity of the models is also a topic of great interest, since, when dealing with very large datasets, we should keep in mind the time that the model will spend training and then predicting each instance. If the model is to be used in daily applications, it must be agile as well as accurate. We now discuss the training time of each model.

Table 6 below shows the training times of each model. It can be seen that, in general, the classical machine learning models would take only a few seconds to train. For the deep learning models, MLP was the fastest, taking only 2.3 min to train and reaching a convergence in only seven epochs.

Another observation that is worth discussing is the great difference in time elapsed from the LSTM and the BERT models; the former took only 8.25 min to converge, while the latter took 196.25 min. Although the F1-score of the BERT model was 72%, which is higher than that of the LSTM model by almost a 6% difference, the elapsed time to train this model was almost 25 times higher. This must be considered when choosing the best model, but it will obviously depend on the kind of application in which it will be used. The model with the highest accuracy/time ratio was MLP.

**Table 6.** Training times per model.

| Model | Time Elapsed (min.) |
|---|---|
| Support Vector Machine | 0.05 |
| Logit Model | 2.5 |
| Gaussian Naive Bayes | 0.04 |
| Multinomial Naive Bayes | 0.025 |
| Random Forest | 4.3 |
| NN-MLP | 2.3 |
| NN-LSTM | 8.25 |
| NN-BERT | 196.91 |

### 3.5. Convergence of the Deep Learning Models

Deep learning models are trained in epochs. An epoch refers to when all the data from the training set are passed through the model [26]. This is usually done in batches of a fixed amount of instances. It must be noted that, normally, a model can learn and achieve better validation accuracy scores by training in every new epoch. However, overfitting must also be considered, as more epochs do not necessarily mean better results. To this end, a percentage of the training set was used as the validation dataset.

Figures 10–12 show that the training and validation loss and accuracy scores converge as the model trains through more epochs. It is interesting to note that both the MLP and LSTM model needed very few epochs to converge, 6 and 5, respectively. In particular, in MLP, we see a divergence of the training and validation very early on.

The BERT model, on the contrary, trains for 12 epochs until the validation and the training scores start to diverge, and early stopping is applied to avoid overfitting.

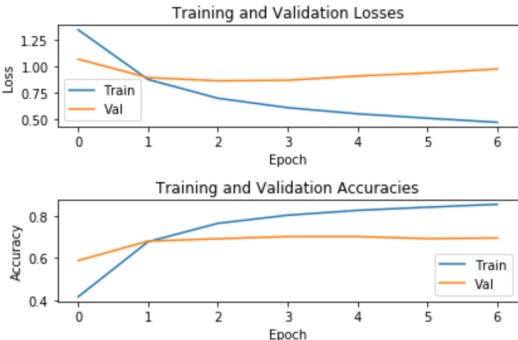

**Figure 10.** MLP—loss and accuracy convergence.

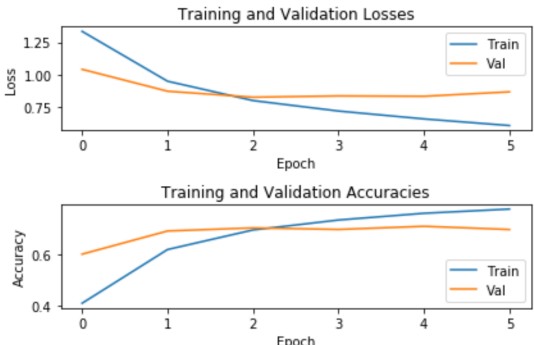

**Figure 11.** LSTM—loss and accuracy convergence.

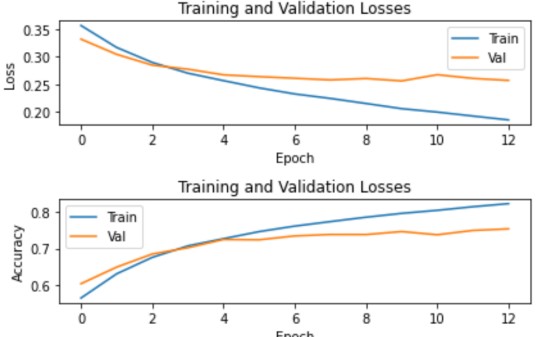

**Figure 12.** BERT—loss and accuracy convergence.

As an example, some of the predictions given by the model are presented in Table 7. It can be seen that, in most cases, the model accurately matches the actual sentiment of the tweet. The two first rows contain a positive and an extremely positive example, which were correctly classified. In both these examples, words such as "sweet", "best" and "nice" were used, which might have made the decision easier. However, in the case of the fourth example, where the model classified the tweet as being negative but it was actually labelled as positive, the tweet did not contain any specific words with positive or negative polarity, and therefore was much more difficult to classify.

**Table 7.** Example tweets with actual and predicted sentiment class.

| Tweet | Actual Sentiment | Predicted Sentiment |
|---|---|---|
| If you re considered at risk during the outbreak now is the time to stock up on supplies including food medicine amp cleaning supplies Be sure you have over the counter medicines amp medical supplies to treat fever amp other symptoms. | Positive | Positive |
| best thing right spread compassion stressed anxious effort nice grocery store clerk reach friend online compliment stranger outfit wash hand | Extremely Positive | Extremely Positive |
| SkinCare: order grocery online standing q master link cooking grocery food household supply skincare | Neutral | Neutral |
| The parade of pharma, and other consumer issues is about the MARKET. Plain and simple. This sociopath does not care about us. #coronapocalypse #Covid19 parade pharma consumer issue market plain simple sociopath care | Negative | Positive |
| remember afford stock pile month food toilet paper sensitive live pay check pay check | Negative | Negative |

## 4. Conclusions

The spread of messages through social media platforms, particularly Twitter, during the COVID-19 pandemic has had a significant impact on almost every user of the platform. Of course, the messages have infinite implications, with some of them being positive, others negative and others being completely neutral. The use of NLP can have a positive impact by identifying and preventing the spread of erroneous, alarmist and hateful messages. Moreover, depression or anxiety problems could be prevented, and some solutions might be implemented to tackle these problems early on, by not allowing them to become out of control.

This study has explored several algorithms for the multi-class classification of tweets related to the COVID-19 pandemic.

The best model was the BERT model, achieving a macro F1-score of 71%, followed by the LSTM model, with an overall macro F1-score of 67%. The deep learning models clearly outperformed the traditional machine learning models.

Future works should be performed by including contextual information such as the date and location of the tweet. These two aspects could give relevant information and more conclusions could be made regarding certain dates and geographical locations around the world.

We also plan to explore other pre-trained language models such as RoBERTa (Robustly Optimized BERT Pretraining Approach) OpenAI's GPT-3, ALBERT or XLNet [27].

**Author Contributions:** Methodology, J.A.J.-V. and I.S.-B.; software, J.A.J.-V.; validation, J.A.J.-V.; formal analysis, J.A.J.-V. and I.S.-B.; investigation, J.A.J.-V. and I.S.-B.; writing J.A.J.-V. and I.S.-B.; supervision, I.S.-B. All authors have read and agreed to the published version of the manuscript.

**Funding:** This work has been supported by the Madrid Government (Comunidad de Madrid) under the Multiannual Agreement with UC3M in the context of "Fostering Young Doctors Research" (NLP4RARE-CM-UC3M), as well as in the context of "Excellence of University Professors"

(EPUC3M17) and in the context of the V PRICIT (Regional Programme of Research and Technological Innovation).

**Institutional Review Board Statement:** Not applicable.

**Informed Consent Statement:** Not applicable.

**Data Availability Statement:** All code used for this study can be found in the following Github repository: https://github.com/Joseantonio-96/Sentiment_Analysis (accessed on 3 November 2021). As for the data used, they can be found at the following Kaggle competition website https://www.kaggle.com/c/sentiment-analysis-of-covid-19-related-tweets/overview/description (accessed on 3 November 2021).

**Conflicts of Interest:** The authors declare no conflict of interest.

## Abbreviations

The following abbreviations are used in this manuscript:

| | |
|---|---|
| NLP | Natural language Processing |
| ML | Machine Learning |
| DL | Deep Learning |
| SVM | Support Vector Machine |
| LSTM | Long Short-Term Memory |
| BERT | Bidirectional Encoder Representation for Transformers |

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
