# Peer review of "Exploring the Impact of COVID-19 on Social Life by Deep Learning"

_information, doi:10.3390/info12110459_

Round 1

Reviewer 1 Report

Good design of the study to address the 'infodemic' issue. Overall, the manuscript has been written well. 

  • It is almost obvious that deep learning would provide better results in most cases. I suggest the authors to make further discussion about the accuracy differences among deep learning models they considered.
  • They need to compare their findings with similar studies/articles from the literature, for example,
    • Satu MS, Khan MI, Mahmud M, et al. Tclustvid: a novel machine learning classification model to investigate topics and sentiment in covid-19 tweets. Knowledge-Based Systems. 2021;226:107126. 
  • The content of figures 7-9 are not visible. Please improve the picture quality of these figures.

Author Response

It is almost obvious that deep learning would provide better results in most cases. I suggest the authors to make further discussion about the accuracy differences among deep learning models they considered.

Response: We have included some sentences to discuss the accuracy differences.

They need to compare their findings with similar studies/articles from the literature, for example, o Satu MS, Khan MI, Mahmud M, et al. Tclustvid: a novel
machine learning classification model to investigate topics and sentiment in covid-19 tweets. Knowledge-Based Systems. 2021;226:107126.

Response: We have included a sentence to reference this work. Unfortunately, we cannot compare it directly to our work because Satu et al. only addressed the classification of negative, neutral and possitive tweets, while we address a finer-grained sentiment analysis of the tweets about the pandemic.

The content of figures 7-9 are not visible. Please improve the picture quality of these figures.

Response: We have improved their quality.

Reviewer 2 Report

The manuscript is to develop Natural Language Processing models using Deep Learning to classify tweets that are related to Covid-19 pandemic.

  1. The contribution of the paper is not very clear. It seems the tested models are just existing approaches. I would suggest the authors summarize the contributions in the introduction section.
  2. It is not clear whether cross-validation has been applied in the study or not. If not, I would suggest the authors add cross-validation in their test since one replication is lack of statistical confidence for authors to trust the results.
  3. Figures 10-12 show that the number of epochs is very small range from 5 to 12 epochs. Are there any particular reasons to select such small epochs?
  4. There are some errors:

Line 75: .. have have..

Line 99-101: there are no citation numbers in the brackets

Line 303: in Figure ??.

Author Response

The contribution of the paper is not very clear. It seems the tested models are just
existing approaches. I would suggest the authors summarize the contributions in the introduction section.

Response: We have clarified this in the introduction. Our main contribution is that we have addressed a finer-grained sentiment analysis of the tweets about the pandemic.

It is not clear whether cross-validation has been applied in the study or not. If not, I would suggest the authors add cross-validation in their test since one
replication is lack of statistical confidence for authors to trust the results.

Response: We have not applied cross-validation due to the high training time for deep learning models. For example, if we applied 10-CV for training BERT, its training time would be multiplied by 10. So, we would need around 32 hours to perform fine-training of the BERT model.

Figures 10-12 show that the number of epochs is very small range from 5 to 12 epochs. Are there any particular reasons to select such small epochs?

Response: We have trained both LSTM and BERT with 20 epochs. However,
Figure 11 shows that LSTM needs very few epochs (5). The BERT model, on the contrary, trains for 12 epochs until the validation and the training scores start to diverge.

There are some errors: Line 75: .. have have..Line 99-101: there are no citation numbers in the brackets Line 303: in Figure ??.

Response: We have fixed them.

Round 2

Reviewer 1 Report

okay, the authors responded to previous comments

Author Response

Thanks for your reviewing.

Reviewer 2 Report

If only one replication is tested, then how could the authors make sure the results really measure the generalization performance of the model. The result might be influenced by a biased test dataset, which could mislead readers. At least, there should be some discussion (literature) to support why one single run and test on the experimental dataset is appropriate. The authors argue that the 10-fold CV is not used because of high computational time. It sounds like not a scientific reason. Weeks running time is common in the machine learning field.
